# The Representation of Females in Studies on Antihypertensive Medication over the Years: A Scoping Review

**DOI:** 10.3390/biomedicines11051435

**Published:** 2023-05-12

**Authors:** Zenab Mohseni-Alsalhi, Maud A. M. Vesseur, Nick Wilmes, Sophie A. J. S. Laven, Daniek A. M. Meijs, Eveline M. van Luik, Esmée W. P. Vaes, Cédric J. R. Dikovec, Jan Wiesenberg, Mohamad F. Almutairi, Emma B. N. J. Janssen, Sander de Haas, Marc E. A. Spaanderman, Chahinda Ghossein-Doha

**Affiliations:** 1Department of Obstetrics and Gynecology, School for Oncology and Developmental Biology (GROW), Maastricht University Medical Center (MUMC+), 6229 ER Maastricht, The Netherlands; 2Faculty of Health, Medicine and Life Sciences (FHML), Maastricht University, 6229 ER Maastricht, The Netherlands; 3Department of Intensive Care Medicine, Maastricht University Medical Center (MUMC+), Maastricht University, 6229 HX Maastricht, The Netherlands; 4Department of Cardiology, Cardiovascular Research Institute Maastricht (CARIM), Maastricht University Medical Center (MUMC+), 6229 ER Maastricht, The Netherlands; 5Department of Obstetrics and Gynecology, Radboud University Medical Center, 6525 GA Nijmegen, The Netherlands

**Keywords:** hypertension, sex stratification, antihypertensive drugs, echocardiography, heart failure

## Abstract

Background: The leading global risk factor for cardiovascular-disease-related morbidity and mortality is hypertension. In the past decade, attention has been paid to increase females’ representation. The aim of this study is to investigate whether the representation of females and presentation of sex-stratified data in studies investigating the effect of antihypertensive drugs has increased over the past decades. Methods: After systematically searching PubMed and Embase for studies evaluating the effect of the five major antihypertensive medication groups until May 2020, a scoping review was performed. The primary outcome was the proportion of included females. The secondary outcome was whether sex stratification was performed. Results: The search resulted in 73,867 articles. After the selection progress, 2046 studies were included for further analysis. These studies included 1,348,172 adults with a mean percentage of females participating of 38.1%. Female participation in antihypertensive studies showed an increase each year by 0.2% (95% CI 0.36–0.52), *p* < 0.01). Only 75 (3.7%) studies performed sex stratification, and this was the highest between 2011 and 2020 (7.2%). Conclusion: Female participation showed a slight increase in the past decade but is still underrepresented compared to males. As data are infrequently sex-stratified, more attention is needed to possible sex-related differences in treatment effects to different antihypertensive compounds.

## 1. Introduction

Cardiovascular diseases (CVDs) are the leading cause of death in females worldwide and responsible for approximately 35.0% of all female deaths in 2019 [1,2,3,4]. Hypertension is the leading risk factor for CVD morbidity and mortality, and is the most substantial and neglected health burden in females [5]. Timely reduction of blood pressure prevents the development of CVD later in life [6]. Historically, females are underrepresented in, or sometimes even excluded from, cardiovascular clinical trials [7,8]. In CVD management, findings from clinical trial data were extrapolated to be equally effective in females and males. Therefore, women are treated equally (the same); however, they are not treated based on equity (i.e., on their health needs). Lack of attention to possible sex-related system-biological differences in (side) effects of pharmacological therapy may have remained undetected and reduced the ability to identify sex-specific differences in important outcomes. If present, this would underscore the need for the development of sex-specific strategies in guideline recommendations for the prevention and management of hypertension and CVD.

The past years, a large number of campaigns led to increased awareness on the impact of CVD in females [9,10,11,12,13,14,15,16]. In 2006, the National Institutes of Health (NIH) decided that drugs are obligated to be tested separately on females but did not require reinvestigation of previously developed drugs. Nonetheless, almost a decade later, important and impactful studies that affected the recommendations in influential guidelines, still not only had sufficient power, nor sex-stratified the findings regarding the female perspective [17,18].

Therefore, we questioned to what extent females were represented for each category of antihypertensive medication and whether the number of included females and sex-stratified data has increased in the past decade compared to the previous decades. We studied the changes in representation of females and sex-stratified data in clinical human studies investigating the five major antihypertensive drug groups.

## 2. Materials and Methods

### 2.1. Scoping Review

This scoping review was conducted according to the Preferred Reporting Items for Systematic Reviews and Meta-analyses (PRISMA) guideline [19]. An extensive systematic literature search was conducted on articles evaluating the effects of antihypertensive medication on cardiovascular and hemodynamic variables. Independent duos of ten investigators (Z.M.-A., M.A.M.V., S.A.J.S.L., E.W.P.V., N.W., D.A.A.M., E.M.v.L., C.J.R.D., J.W. and M.F.A.) systematically searched PubMed (NCBI) and Embase (Ovid) databases from inception until May 2020. They reviewed original research articles investigating antihypertensive medication studies over the years (for keywords used in the literature search, see Table 1).

The search, inclusion, and exclusion criteria were initially developed for a series of systematic reviews and meta-analyses to assess the effect of the five major groups of antihypertensive drugs on cardiovascular outcomes in females specifically, as compared to males. The current scoping review preludes these meta-analyses and primarily investigates the change over time in the proportion of included females in the studies. Secondary, it investigates to what extent sex stratification was performed.

### 2.2. Selection of Studies

The identified articles were assessed for eligibility in two phases (Figure 1). First, individual studies were screened based on the title and abstract by independent duos of ten investigators. A predefined study-extraction sheet was used. Second, articles were screened based on full-text suitability based on inclusion and exclusion criteria also by independent duos of the same ten investigators (title–abstract pairs: M.F.A.–E.W.P.V., C.J.R.D.–S.A.J.S.L., E.M.v.L.–D.A.M.M., Z.M.-A.–J.W., M.A.M.V.–N.W.; full-text pairs: C.J.R.D.–N.W., E.M.v.L.–M.A.M.V., D.A.M.M.–S.A.J.S.L., E.W.P.V.–J.W.). Discrepancies for the first and second selection were resolved by mutual agreement of two investigators. When no conclusion could be made, a third investigator was involved to make the decision.

If studies: (1) investigated one class of the five main groups of antihypertensives (beta-blockers (BB), angiotensin-converting enzyme inhibitors (ACEI), angiotensin receptor blockers (ARB), calcium channel blockers (CCB), and diuretics (DIU, all different types could be included), (2) human studies, (3) adults ≥ 18 years of age, and (4) articles written in English or Dutch, then they were included.

Only primary studies were used, so studies with re-usage of a cohort for secondary analysis were excluded. Articles were also excluded if: (1) only the abstract was available and the full report was not obtainable, (2) unsuitable study design (systematic reviews and meta-analyses, literature reviews, case reports, animal studies, and in vitro studies), (3) no reference group included (control, placebo, or other antihypertensive medication group), (4) outcome not related to cardiovascular health, (5) no information regarding specific dose and duration.

### 2.3. Data Extraction

A predefined data-extraction sheet, including characteristics and outcomes of interest, was used. Study characteristics were extracted, including sample size, sex-stratified outcomes (yes/no), total number of males and total number of females, number of different medication types, names and groups of the different medication types, type of study, and duration of follow-up in days. Anthropometric measures and effect measures, with standard deviation (SD), standard error (SE), or 95% confidence interval (CI), were collected from the eligible studies in predesigned data collection forms. The potential effect of antihypertensives was beyond the scope of this review. Extractions were performed independently by two investigators and then compared to each other to ensure quality and reliability (data extraction pairs: Z.M.–E.V., C.D.–S.L., E.v.L.–D.M., M.V.–N.W.). Discrepancies were resolved by dialogue or discussion with a third independent investigator. Google sheets were used to streamline the data extraction process.

### 2.4. Statistical Analysis

Statistical analyses were performed using SPSS Statistics version 27·0 (IBM Corp., Armonk, NY, USA). Baseline characteristics and normally distributed variables were reported as mean ± SD. Categorical variables were expressed by number (percentage). Linear regression analyses with beta coefficient (β) and 95% CI were performed to explore the associations between the percentage of females included in the studies. The R-squared provided the amount of variability in the outcome that was accounted for by the predictor variable. We considered a *p*-value below 0.05 to be statistically significant. Statistical analyses were performed by four investigators (M.V., Z.M., N.W., E.V.).

## 3. Results

### 3.1. Study Selection

The search strategy resulted in 73,867 potential articles. Based on title and abstract, 58,737 records were excluded resulting in a total of 15,130 studies eligible for full-text screening. Based on the full text, 13,051 were additionally excluded articles due to the following reasons: article unfindable (*n* = 766), unsuitable study design (*n* = 6226), no reference value measurement (*n* = 153), outcome not related to CV health (*n* = 3864), data not suitably reported (*n* = 535), no registration of dose or duration of the medication (*n* = 984), and the presence of other rare non-hemodynamic comorbidities that may hamper generalization of the effect of drug therapy (i.e., concomitant use of biologicals, specific genetic translocation, chemotherapy, and kidney transplantation) (*n* = 523). Eventually, after excluding all unsuitable articles, 2046 articles were eligible for data extraction (Figure 1).

### 3.2. Study Characteristics

Out of the 2046 studies, 1198 (58.6%) studies were randomized controlled trials (RCTs), 773 (37.8%) prospective cohort studies, 43 (2.1%) retrospective cohort studies, 22 (1.1%) case–control studies, and four were cross-sectional studies (0.2%). (Table 2). The remaining six (0.3%) studies had another study design, including multicenter open-label studies (*n* = 2), open-label trials (*n* = 2), open non-randomized preliminary study (*n* = 1), and a pilot study (*n* = 1) (Table 2, Appendix A).

### 3.3. Prevalence of Included Females and Sex-Stratified Data

The total number of participants in all studies was 1,348,172 adults, of which 38.1% were females, 60.3% males, and 1.6% did not mention sex as seen in Table 3. Sex distinction was not mentioned in 126 (6.2%) studies. Sex-stratified outcome data was not shown in 1706 (83.4%) studies. Female-only cohorts were used in eight (0.4%) studies and male-only cohorts in 131 (6.2%) studies. Sex stratification was performed in 75 (3.7%) studies.

Figure 2 shows the cumulative sum of included females and males over the years from 1964 to 2020. Linear regression analysis revealed that the percentage of females participating in antihypertensive studies increased by 0.2% (95% CI 0.36–0.52%) per year (Table 3). The yearly increase was highest between 2001 and 2010 being 0·5% (95% CI 0.08–0.95) (Table 3). Despite the increase in included females and sex-stratified data, between 2011 and 2020, 38.7% of included participants were female and 7.2% showed sex-stratified data. 

### 3.4. Prevalence of Females and Sex-Stratified Data Selective Antihypertensive Medication

Our final analysis is based on 2046 unique antihypertensive papers ranging from 1964 to 2020 studying the effect of 2341 (100.0%) different antihypertensive medicament cohorts (Table 3). Of all studies, 252 (10.8%) studies reported data on DIU, 652 (27.9%) on BB, 442 (18.9%) on CCB, 671 (28.7%) on ACEI, and 324 (13.8%) on ARB.

### 3.5. Diuretics

In total, 252 studies reported data on DIU between 1964 and 2019. The number of females and males separately was not mentioned in 13 (5.2%) studies. Sex-stratified outcomes were not shown in two hundred and fourteen (84.9%) studies, female-only cohorts were reported in two (0.8%) studies, and male-only cohorts in thirteen (5.2%) studies. Sex stratification in primary study analysis was performed in ten (4.0%) studies. The total number of participants in all studies was 389,873 adults, of which 44.2% were female and 55.5% male. Figure 3 shows the cumulative sum of included females and males over the years from 1964 to 2019 as seen in Table 4. Between 2011 and 2020, 6177 (25.3%) of the included participants were females, and five (7.5%) studies reported sex-stratified analyses.

### 3.6. Beta-Blockers

In total, 652 studies reported data on BB between 1968 and 2020. The number of females and males separately was not mentioned in 46 (7.1%) studies. Sex-stratified data was not shown in five hundred and thirty-three (81.7%) studies, female-only cohorts were reported in three (0.5%) studies and male-only cohorts in fifty-one (7.8%) studies. Sex stratification was performed in 19 (2.9%) studies. The total number of participants in all the studies was 340,718, of which 36.6% were females, 57.8% males, and 5.6% had undefined sex. Figure 4 shows the cumulative sum of included females and males over the years from 1968 to 2020. The number of females included in the studies increased slightly over the years as seen in Table 5. Between 2011 and 2020, less than half, 16,061 (36.7%) of included participants were females, and five (5.2%) studies reported sex-stratified data.

### 3.7. Calcium Channel Blockers

In total, 442 studies reported data on CCB between 1977 and 2020. The number of females and males separately was not mentioned in 28 (6.3%) studies. Sex-stratified data was not shown in three hundred and sixty-eight (83.3%) studies, female-only cohorts were reported in two (0.5%) studies, and male-only cohorts in thirty (6.8%) studies. Sex stratification was performed in 14 (3.2%) studies. The total number of participants in all studies was 464,084, of which 41.9% were females, 57.7% males, and 0.4% were undefined. Figure 5 shows the cumulative sum of included females and males over the years from 1977 to 2020. The number of females included in the studies increased slightly over the years as seen in Table 6. Between 2011 and 2020, almost half, 11,234 (42.8%) of included participants were females, and four (10.3%) studies reported sex-stratified data.

### 3.8. Angiotensin-Converting Enzyme Inhibitors

In total, 671 studies reported data on ACEI between 1979 and 2019. The number of females and males separately was not mentioned in 31 (4.6%) studies. Sex-stratified data was not shown in 564 (84.1%) studies, female-only cohorts were reported in 0 (0.0%) studies, and male-only cohorts in 31 (4.6%) studies. Sex stratification was performed in 28 (4.2%) studies. The total number of participants in all studies was 546,164, of which 36.9% were females, 62.6% males, and 0.5% were undefined. Figure 6 shows the cumulative sum of included females and males over the years from 1979 to 2019. The number of females included in the studies increased slightly over the years as seen in Table 7. Between 2011 and 2020, still only 14,577 (32.3%) of included participants were females, and seven (20.6%) studies reported sex-stratified data.

### 3.9. Angiotensin Receptor Blockers

In total, 324 studies reported data on ARB between 1984 to 2020. The number of females and males separately was not mentioned in eight (2.5%) studies. Sex-stratified data was not shown in 295 (91.0%) studies, female-only cohorts were reported in zero (0.0%) studies, and male-only cohorts in seven (2.2%) studies. Sex stratification was performed in 14 (4.3%) studies. The total number of participants in all studies was 379,302, of which 42.2% were females, 55.3% males, and 2.5% were undefined. Figure 7 shows the cumulative sum of included females and males over the years from 1984 to 2020. The number of females included in the studies increased slightly over the years as seen in Table 8. Between 2011 and 2020, almost half, 35,231 (46.2%) of included participants were females, and four (4.8%) studies reported sex-stratified data.

## 4. Discussions

In the current scoping review, 2046 unique antihypertensive studies comprising 1,395,264 individuals were analyzed over a period between 1964 and 2020. We observed that females are still substantially underrepresented in clinical trials. Despite a yearly increase of 0.2% in included females, in the last decade, one third of the included participants were female. Regarding subtypes of antihypertensive medications, studies on BB least included females and studies on DIU included the highest number of females. Sex stratification in primary study analysis was performed in 3.7% of all included studies over the years, with a percentage of 7.2% in the last decade. 

Hypertension is the leading risk factor for stroke, arrhythmia, coronary heart disease, acute coronary syndrome, and heart failure. In line with previous findings, we observed that studies on antihypertensive treatment using the five major antihypertensive drug groups to lower the risk on these CVD outcomes, included less females than males [20,21,22]. As we observed a gradual increase in included women and the sex-stratified display of studied findings, the current awareness of possible differences in effectiveness of antihypertensive compounds is likely to result in a balanced female-to-male inclusion ratio or sex-stratified reports; however, given the slow speed of change, attention is still needed.

Melloni et al. [23] studied 156 RCT’s cited by the 2007 females’ prevention guidelines to determine female representation over time. Considering only trials that enrolled both sexes, female enrolment was 18.0% in 1970, and increased to 34.0% in 2006. They also found that female representation was higher in international versus United-States-only trials (32.7% versus 26.7%) and in primary versus secondary prevention trials (42.6% versus 26.6%). Representation of females was highest among trials in hypertension (44.0%), diabetes (40.0%), and stroke (38.0%), and lowest for heart failure (29.0%), coronary artery disease (25.0%), and hyperlipidemia (28.0%).^23^ This underrepresentation of females in RCTs and the lack of reporting sex-specific results may imply that the guidelines for CVD prevention in females cannot be reliably based on these studies. In this line of reasoning, cardiovascular risk stratification in this population will need to be established such that risk stratifications are not extrapolated from largely male-retrieved evidence.

Sex-differences in response to antihypertensive treatment may originate from differences in pharmacodynamics, pharmacokinetics, pharmacogenomics, and differences in distribution space. In addition, concurrent disease, originating from different underlying sex-sensitive pathology, may also affect altered responsiveness. These differences may underlie altered responses, side effects, and prognosis between females and males to comparable dosages of antihypertensives [24,25,26,27]. Females report more drug-related side effects from antihypertensive therapy than males [28]. Bots et al. demonstrated that females report more adverse drug reactions and different types of adverse drug reactions compared with men when using ACEI [29]. The sex-specific mechanisms contributing to the multifactorial pathogenesis and varying consequences of hypertension in females are only partly explained in most studies. In addition to the mentioned handling and distribution differences, differences in blood pressure set points and consequently blood pressure targets may also relate to the effectiveness, safety, and gained health [30].

Ljungman et al. described sex differences in antihypertensive treatments and investigated if this could be due to comorbidities [31]. They performed a cohort study within the Swedish Primary Care Cardiovascular Database (40,825 individuals). They found that females were more often treated with diuretics and men with ACEI. Comorbidities could not entirely explain sex differences in antihypertensive treatment using regression models. Men were prescribed ACEI/ARBs and interrupted treatment more often than females. Leading to the conclusion that females and males were treated differently using antihypertensive medications in blood pressure control and they cannot fully explain this by the difference in comorbidities. Walletin et al. investigated the antihypertensive drug treatment in Sweden using the Stockholm Regional Healthcare Data Warehouse (292,428 individuals) [32]. Females more often used DIU, ARB, and BB, while males used more ACEI and CCB. A total of 66% of the females with diabetes and 72% with heart failure used ACEI/ARB versus 76% and 79% in men, respectively. This shows that comorbidities are taken into consideration for antihypertensive medication treatments. However, sex differences in antihypertensive drug treatments prevail.

The lower participation rate of females in studies on antihypertensive drugs raises the question why females are underrepresented. Historically, as a result of excluding women of childbearing age from clinical trials for various reasons, knowledge about diseases originated mainly from research on male animals, cells, and humans [33]. Despite the fact that multiple policies in Canada, the USA, and Europe were implemented to stimulate researchers to take sex and gender into account, it is still viewed as a specialized field of interest rather than a central condition in medical research [34]. In addition, the enrolment process in a study is multifactorial involving several investigators and patient-based and community-based factors. First, the investigator has to incorporate the requirement of balanced participation of both sexes in study designs, unless there is a special need for a study with one sex only. Second, females must actively be made aware of the opportunity to participate via consumer channels. Attention may be needed to the way information is provided as females tend to make decisions differently compared to males, which could lead to different enrolment rates by sex. Generally, females take more time to make a decision, and they require more sources of background information with substantiated pros and cons [35]. Females’ participation may be enhanced with sex-specific clinical trial education materials [22]. From a patient’s perspective, the willingness to participate may also vary between sexes. Ding et al. conducted a randomized study about patient willingness to participate in cardiovascular prevention trials and concluded that males had 15.0% greater willingness to participate compared to females [36]. Among the reasons for this difference was that females generally perceived a greater risk of harm from trial participation. Females had also been shown to take fewer risks than males under stress conditions, and important health-based decisions can be considered a source of stress [37]. RCTs present an added element of risk and uncertainty, and females have been shown to be more reluctant than males to consider participation [38]. From a community perspective, first, lack of support to participate, because of logistic problems such as transportation and childcare, may obstruct the decision to participate. Second, in communities with low literacy, females might have less possibilities to understand information brochures to be comfortable with the clinical trial process, with the process of informed consent, and with the overall clinical trial experience.

### 4.1. Clinical Implications and Recommendations for Future Research

A useful first step is to communicate better the vast knowledge on sex differences in hypertension management. To ensure meaningfully obtained sex-specific results, novel trial design solutions may be needed. Potential barriers need to be overcome to increase female trial participation, including enhancing the understanding of trial process and logistics, but also enhancing limited clinical trial information from physicians, and misperceptions around the risks and benefits of participation. Furthermore, study protocols should facilitate comparable representation of both sexes by adequate sex-specific powering of trial participants to ensure a large enough group of each sex and by presenting data in a sex-stratified assessment. 

Although these designs could be more costly, trial costs can be reduced by specifying the power to detect the clinically relevant interaction effect between sex and treatment, recruiting a prespecified number of females and males, and ceasing enrolment of the particular sex when the sample size target is reached. For already published studies, investigators might consider performing post hoc sex-specific analyses, while considering potential power insufficiencies in the interpretation of the results [39]. Generating an open access database with data from completed clinical trials may be helpful to optimize power allowing individual patient data analysis.

### 4.2. Strengths and Limitations

This scoping review includes a substantial number of studies in a systematic way aiming to include all relevant studies which resulted in an analysis of the lack of sex-specific outcome measures in antihypertensive studies. Additionally, we formed applicable recommendations to help improve further research not only in the field of antihypertensive treatment but also beyond this scope.

There are some limitations that need to be addressed. First, our study cannot provide insights into the reasons for low female participation in hypertension treatment trials. Nonetheless, our study provides consistent information regarding the still present underrepresentation of females in antihypertensive trials. Moreover, reported data also might still benefit from reporting sex-stratified data. Furthermore, our baseline characteristics were not separated based on sex, despite this being our main point of interest. Due to only such a small number of articles separating their outcomes based on sex or only including females, it was not possible to divide the baseline characteristics on a sex-based manner. However, as we merely investigated the representation of females as compared to males in antihypertensive trials, we believe this has not affected our findings and that we still addressed the gap in sex-stratified antihypertensive treatment data.

## 5. Conclusions

In the currently available clinical studies on pharmacological antihypertensive treatment, females are still underrepresented in the most used antihypertensive drug modalities (less than 40.0%). Despite the significant yearly increase in reported sex-stratified data over the past decades, as this only occurs in less than 10.0% of studies, attention to sex stratification and representation in future antihypertensive studies is still needed.

## Figures and Tables

**Figure 1 biomedicines-11-01435-f001:**
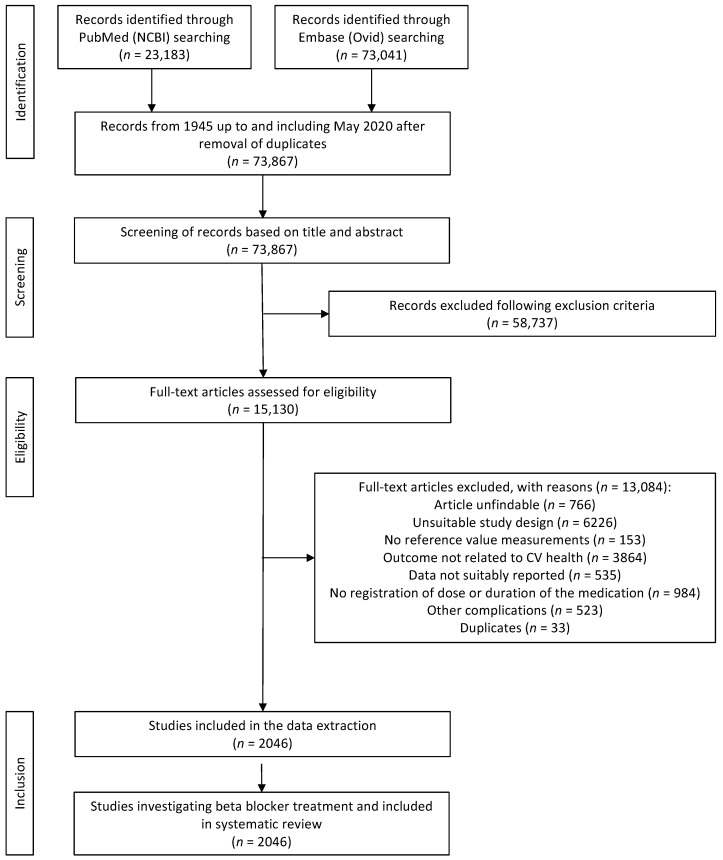
Flowchart summarizing the process of study selection.

**Figure 2 biomedicines-11-01435-f002:**
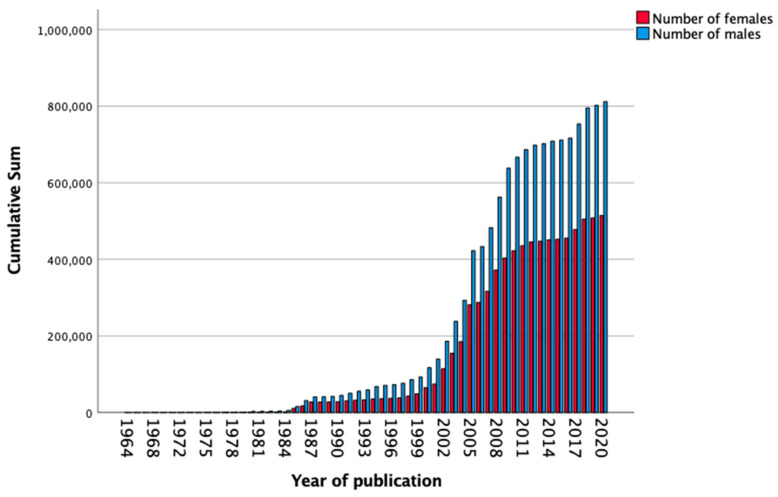
Cumulative sum of the number of females and males (*n*) participating in included studies over the years from 1964 to 2020.

**Figure 3 biomedicines-11-01435-f003:**
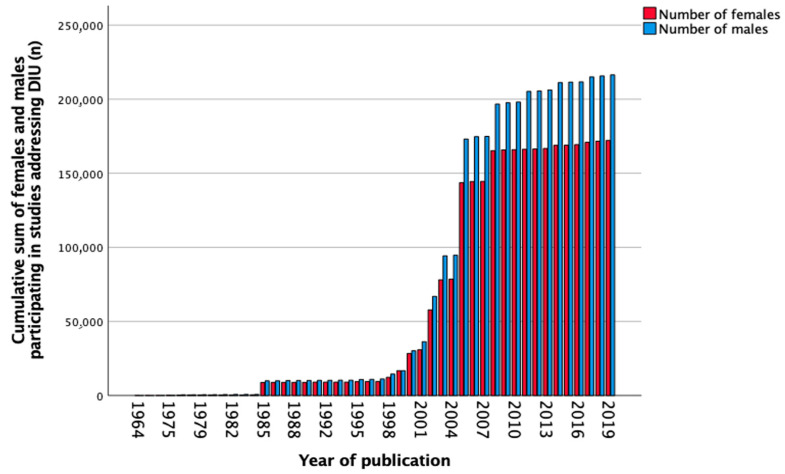
Cumulative sum of the number of females and males (*n*) participating in studies addressing diuretics (DIU) over the years.

**Figure 4 biomedicines-11-01435-f004:**
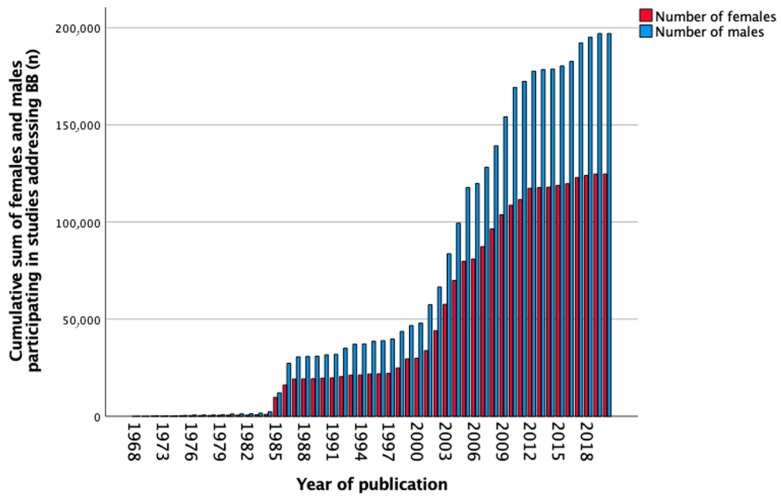
Cumulative sum of the number of females and males (*n*) participating in studies addressing beta-blockers (BB) over the years.

**Figure 5 biomedicines-11-01435-f005:**
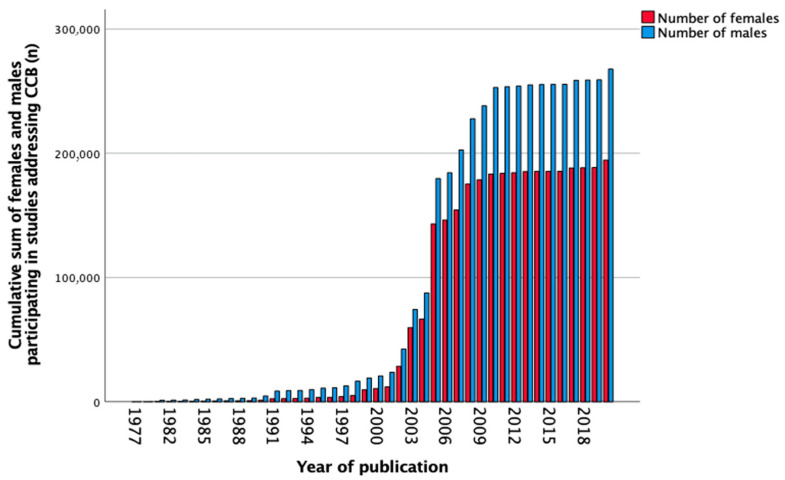
Cumulative sum of the number of females and males (*n*) participating in studies addressing calcium channel blockers (CCB) over the years.

**Figure 6 biomedicines-11-01435-f006:**
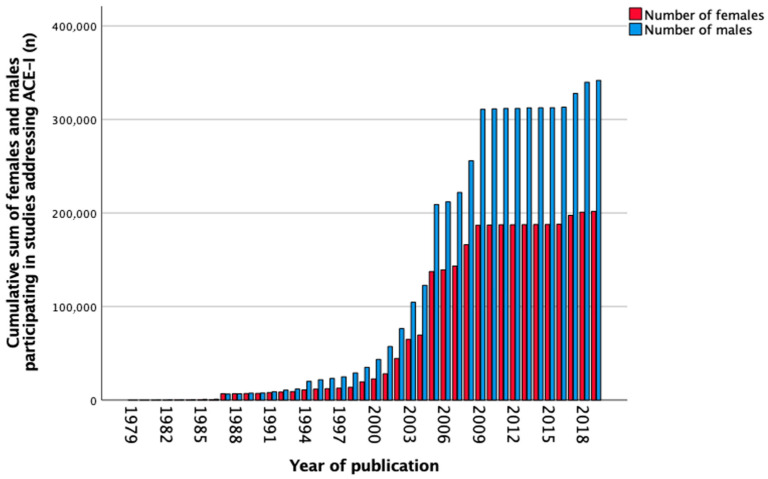
Cumulative sum of the number of females and males (*n*) participating in studies addressing angiotensin-converting enzyme inhibitors (ACEI) over the years.

**Figure 7 biomedicines-11-01435-f007:**
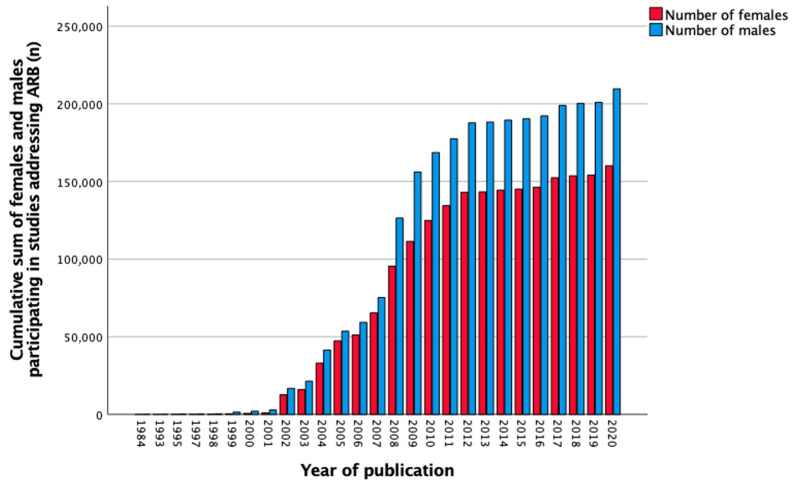
Cumulative sum of the number of females and males (*n*) participating in studies addressing angiotensin receptor blockers (ARB) over the years.

**Table 1 biomedicines-11-01435-t001:** Literature search strategy for PubMed (NCBI) and Embase (Ovid) databases.

PubMed	Embase
Component 1: Antihypertensive medication: “diuretics” OR “adrenergic beta-antagonists” OR “beta blockers” [Title/Abstract] OR “Antihypertensive agents” OR “blood pressure lowering therapy” [Title/Abstract] OR “antihypertensive medication” [Title/Abstract] OR “antihypertensive therapy” [Title/Abstract] OR “angiotensin-converting enzyme inhibitors” OR “ACE inhibitors” [Title/Abstract] OR “Angiotensin receptor antagonists” OR “angiotensin receptor blockers” [Title/Abstract] OR “sympatholytics” OR “Calcium Channel Blockers”	Component 1: Antihypertensive medication: exp diuretic agent/or exp beta adrenergic receptor blocking agent/or exp adrenergic receptor blocking agent/or exp antihypertensive agent/or exp dipeptidyl carboxypeptidase inhibitor/or exp angiotensin receptor antagonist/or exp calcium channel blocking agent.ti,ab.
Component 2: Cardiac geometry: “ventricular remodeling” OR “ventricular remodeling” [Title/Abstract] OR “cardiac remodeling” [Title/Abstract] OR “cardiac adaptation” [Title/Abstract] OR “LV geometry” [Title/Abstract] OR “left ventricular geometry” [Title/Abstract] OR “cardiac geometry” [Title/Abstract] OR “cardiac dimension” [Title/Abstract] OR “left ventricle remodeling” [Title/Abstract] OR “Hypertrophy, Left Ventricular” OR “left ventricular hypertrophy” [Title/Abstract] OR “echocardiography” OR Echocardiography [Title/Abstract] OR “left ventricular mass” [Title/Abstract] OR “left ventricular mass index” [Title/Abstract] OR “relative wall thickness” [Title/Abstract] OR “concentric cardiac remodeling” [Title/Abstract] OR “eccentric cardiac remodeling” [Title/Abstract]	Component 2: Cardiac geometry: exp heart ventricle remodeling/or (ventricular remodeling or cardiac remodeling or cardiac adaptation or LV geometry or left ventricular remodeling or cardiac geometry or cardiac dimension).ti,ab. or exp echocardiography/ or echocardiography.ti,ab.
Component 3: Heart failure: “Heart Failure” OR “Heart Failure, Systolic”	Component 3: Heart failure: exp heart failure.ti,ab.
Component 4: Diastolic dysfunction: “heart failure, diastolic” OR “diastolic dysfunction” [Title/Abstract]	Component 4: Diastolic dysfunction: exp diastolic dysfunction/or diastolic function.ti,ab.
Component 5: Myocardial infarction: “myocardial infarction” OR “myocardial infarction” [Title/Abstract] OR “acute myocardial infarction” [Title/Abstract] OR “heart attack” [Title/Abstract]	Component 5: Myocardial infarction: exp heart infarction.ti,ab.
Component 6: CVA: Stroke OR “cerebrovascular accident” [Title/Abstract] OR “acute cerebrovascular accident” [Title/Abstract] OR “acute cerebrovascular insult” [Title/Abstract]	Component 6: CVA: exp cerebrovascular accident.ti,ab.

CVA: cardiovascular accident, ACE: angiotensin-converting enzyme, LV: left ventricle.

**Table 2 biomedicines-11-01435-t002:** Overall characteristics of the included antihypertensive studies (*n* = 2046) in the scoping review.

Number of Participants *n* (%)	*n* = 2046
Total	1,348,172 (100%)
Females	514,604 (38.2%)
Males	812,397 (60.3%)
Unspecified	21,171 (1.6%)
Mean age ± SD	58.0 ± 8.8
Sex stratification *n* (%)	
Not stratified	1706 (83.4%)
Stratified	75 (3.7%)
Only females	8 (0.4%)
Only males	131 (6.4%)
Not mentioned	126 (6.2%)
Study design *n* (%)	
Randomized controlled trial	1198 (58.6%)
Prospective cohort study	737 (37.8%)
Retrospective cohort study	43(2.1%)
Case–control study	22 (1.1%)
Cross-sectional study	4 (0.2%)
Other	6 (0.3%)
Effect *n* (%)	
Acute	419 (20.5%)
Chronic	1627 (79.5%)

**Table 3 biomedicines-11-01435-t003:** Characteristics of the included antihypertensive studies in the scoping review divided for each decade.

	Total	1964–1980	1981–1990	1991–2000	2001–2010	2011–2020
Number of studies						
Total, *n* (%)	2046 (100)	58 (2.8)	420 (20.5)	544 (26.6)	690 (33.7)	334 (16.3)
Without mentioning sex distinction, *n* (%)	126 (6.2)	5 (8.6)	60 (14.3)	31 (5.7)	21 (3.0)	9 (2.7)
Without sex stratification, *n* (%)	1706 (83.4)	36 (62.1)	281 (66.9)	466 (85.7)	629 (91.2)	294 (88.0)
With sex stratification, *n* (%)	75 (3.7)	3 (5.2)	14 (3.3)	14 (2.6)	20 (2.9)	24 (7.2)
Only including females, *n* (%)	8 (0.4)	0 (0.0)	0 (0.0)	3 (0.6)	1 (0·1)	4 (1·2)
Only including males, *n* (%)	131 (6.4)	14 (24.1)	65 (15.5)	31(5.7)	19 (2.8)	3 (0.9)
Participants						
Total, *n* (%)	1,348,172 (100)	4636 (0.3)	77,070 (5.7)	111,616 (8.3)	917,034 (68.0)	237,816 (17.6)
Females, *n* (%)	514,604 (38.2)	1132 (24.4)	27,347 (35.5)	36,423 (32.6)	357,636 (39.0)	92,066 (38.7)
Males, *n* (%)	812,397 (60.3)	3033 (65.4)	41,765 (54.2)	72,549 (65.0)	549,607 (59.9)	145,443 (61.1)
Sex not known, *n* (%)	21,171 (1.6)	471 (10.2)	7958 (10.3)	2644 (2.4)	9791 (1.1)	332(0.1)
Trends						
Trend of increase in females per year	β = 0.9 (95% CI 0.361–0.517, *p* < 0.001 *)	β = −1.342 (95% CI −2.752–0.068, *p* = 0.062)	β = 0.449 (95% CI −0.368–1.266, *p* = 0.280)	β = 0.229 (95% CI −0.515–0.973, *p* = 0.545)	β = 0.515 (95% CI 0.076–0.954, *p* = 0.022 *)	β = 0.274 (95% CI −0.501–1.049, *p* = 0.487)
Trend of increase in females in studies stratifying based on sex	β = 0.517 (95% CI 0.064–0.969, *p* = 0.026 *)	β = −2.887 (95% CI −27.937–22.162, *p* = 0.381)	β = −2.167 (95% CI −7.567–3.234, *p* = 0.399)	β = −2.926 (95% CI −7.454–1.601, *p* = 0.183)	β = 2.496 (95% CI −0.520–5.511, *p* = 0.099)	β = 0.570 (95% CI −4.818–5.958, *p* = 0.828)
Age						
Mean age (range, SD)	58.0 (22–86, 8.8)	53.6 (39–72, 6.4)	54.7 (22–82, 8.7)	56.6 (22–81, 8.9)	59.5 (22–86, 8.4)	61.1 (25–86, 9.1)
Medication groups						
Diuretics, *n* (%)	252 (10.8)	11 (23.4)	37 (8.9)	48 (7.2)	89 (10.0)	67 (20.9)
Beta-blockers, *n* (%)	652 (27.9)	25 (53.2)	120 (28.8)	146 (22.0)	265 (29.7)	96 (30.9)
Calcium channel blockers, *n* (%)	442 (18.9)	4 (8.5)	128 (30.7)	149 (22.4)	122 (13.7)	39 (12.2)
Angiotensin-converting enzyme inhibitors, *n* (%)	671 (28.7)	7 (14.9)	130 (31.2)	294 (44.2)	206 (23.1)	34 (10.6)
Angiotensin receptor blockers, *n* (%)	324 (13.8)	0 (0.0)	2 (0.5)	28 (4.2)	210 (23.5)	84 (26.3)

Categorical variables are presented as number (%). Continuous variables are presented as mean (range, standard deviation). Valid percentages are presented. Linear regression analysis provides beta, the 95% confidence interval, and the *p*-value. All *p*-values < 0.05 were considered statistically significant and star-truncated.

**Table 4 biomedicines-11-01435-t004:** Characteristics of the included diuretics (DIU) studies in the scoping review.

	Total	1964–1980	1981–1990	1991–2000	2001–2010	2011–2020
Number of studies						
Total, *n* (%)	252 (100)	11 (4.4)	37 (14.7)	48 (19.0)	89 (35.3)	67 (26.6)
Without mentioning sex distinction, *n* (%)	13 (5.2)	0 (0.0)	5 (13.5)	4 (8.3)	3 (3.4)	1 (1.5)
Without sex stratification, *n* (%)	214 (84.9)	9 (81.8)	26 (70.3)	40 (83.3)	81 (91.0)	58 (86.6)
With sex stratification, *n* (%)	10 (4.0)	1 (9.1)	0 (0.0)	1 (2.1)	3 (3.4)	5 (7.5)
Only including females, *n* (%)	2 (0.8)	0 (0.0)	0 (0.0)	1 (2.1)	0 (0.0)	1 (1.5)
Only including males, *n* (%)	13 (5.2)	1 (9.1)	6 (16.2)	2 (4.2)	2 (2.2)	2 (3.0)
Participants						
Total, *n* (%)	389,873 (100)	956 (0.2)	18,349 (4.7)	40,554 (10.4)	305,510 (78.4)	24,504 (6.3)
Females, *n* (%)	172,154 (44.2)	385 (40.0)	8621 (47.0)	19,429 (47.9)	137,542 (45.0)	6177 (25.3)
Males, *n* (%)	216,402 (55.5)	572 (59.8)	9683 (52.8)	19,944 (49.2)	167,892 (55.0)	18,311 (74.7)
Sex not known, *n* (%)	1317 (0.3)	0 (0.0)	45 (0.2)	1181 (2.9)	76 (0.0)	16 (0.0)
Total participants in studies that stratified sex, *n* (%)	8259 (100)	53 (0.6)	128 (1.5)	1155 (14.0)	268 (3.2)	6655 (80.7)
Included females in studies that stratified sex, *n* (%)	253 (3.1)	22 (41.5)	0 (0.0)	18 (1.6)	107 (40.0)	106 (1.6)
Included males in studies that stratified sex, *n* (%)	8006 (96.9.)	31 (58.5)	12 (100.0)	1137 (98.4)	161 (60.0)	6549 (98.4)
Age						
Mean age, (range, SD)	60.8 (27–86, 9.0)	50.1 (39–59, 9.6)	56.8 (40–78, 8.7)	58.6 (27–76, 7.4)	61.6 (27–86, 9.2)	64.3 (41–78, 10.0)

Categorical variables are presented as number (%). Continuous variables are presented as mean (range, standard deviation). Valid percentages are presented.

**Table 5 biomedicines-11-01435-t005:** Characteristics of the included beta-blockers (BB) studies in the scoping review.

	Total	1968–1980	1981–1990	1991–2000	2001–2010	2011–2020
Number of studies						
Total, *n* (%)	652 (100)	25 (3.8)	120 (18.4)	146 (22.4)	265 (40.6)	96 (14.7)
Without mentioning sex distinction, *n* (%)	46 (7.1)	3 (12.0)	14 (11.7)	13 (8.9)	14 (5.3)	2 (2.1)
Without sex stratification, *n* (%)	533 (81.7)	17 (68.0)	78 (65.0)	117 (80.1)	234 (88.3)	87 (90.6)
With sex stratification, *n* (%)	19 (2.9)	0 (0.0)	3 (2.5)	2 (1.4)	9 (3.4)	5 (5.2)
Only including females, *n* (%)	3 (0.5)	0 (0.0)	0 (0.0)	1 (0.7)	1 (0.4)	1 (1.0)
Only including males, *n* (%)	51 (7.8)	5 (20.0)	25 (20.8)	13 (8.9)	7 (2.6)	1 (1.0)
Participants						
Total, *n* (%)	340,718 (100)	2211 (0.6)	56,493 (16.6)	28,568 (8.4)	209,644 (61.5)	43,802 (12.9)
Females, *n* (%)	124,711 (36.6)	586 (26.5)	19,060 (33.7)	10,253 (35.9)	78,751 (37.6)	16,061 (36.7)
Males, *n* (%)	197,044 (57.8)	1166 (52.7)	30,475 (53.9)	16,340 (57.2)	121,322 (57.9)	27,741 (63.3)
Sex not known, *n* (%)	18,963 (5.6)	459 (20.8)	6958 (12.3)	1975 (6.9)	9571 (4.5)	0 (0.0)
Total participants in studies that stratified sex, *n* (%)	13,460 (100)	103 (0.8)	497 (3.7)	1745 (13.0)	10,640 (79.0)	484 (3.6)
Included females in studies that stratified sex, *n* (%)	5835 (43.4)	0 (0.0)	5 (1.0)	31 (1.8)	5511 (51.8)	288 (59.5)
Included males in studies that stratified sex, *n* (%)	7625 (56.6)	103 (100)	492 (99.0)	1714(98.2)	5129 (48.2)	196 (40.5)
Age						
Mean age (range, SD)	56.3 (22–77, 8.6)	53.6 (41–72, 7.7)	51.2 (22–76, 8.5)	55.3 (22–77, 7.6)	58.1 (22–76, 8.8)	58.5 (25–77, 9.8)

Categorical variables are presented as number (%). Continuous variables are presented as mean (range, standard deviation). Valid percentages are presented.

**Table 6 biomedicines-11-01435-t006:** Characteristics of the included calcium channel blockers (CCB) studies in the scoping review.

	Total	1977–1980	1981–1990	1991–2000	2001–2010	2011–2020
Number of studies						
Total, *n* (%)	442 (100)	4 (0.9)	128 (29.0)	149 (33.7)	122 (27.6)	39 (8.8)
Without mentioning sex distinction, *n* (%)	28 (6.3)	1 (25.0)	16 (12.5)	9 (6.0)	1 (0.8)	1 (2.6)
Without sex stratification, *n* (%)	368 (83.3)	3 (75.0)	86 (67.2)	128 (85.9)	119 (97.5)	32 (82.1)
With sex stratification, *n* (%)	14 (3.2)	0 (0.0)	6 (4.7)	2 (1.3)	2 (1.6)	4 (10.3)
Only including females, *n* (%)	2 (0.5)	0 (0.0)	0 (0.0)	0 (0.0)	0 (0.0)	2 (5.1)
Only including males, *n* (%)	30 (6.8)	0 (0.0)	20 (15.6)	10 (6.7)	0 (0.0)	0 (0.0)
Participants						
Total, *n* (%)	464,084 (100)	1514 (0.3)	4830 (1.0)	26699 (5.8)	404,807 (87.2)	26234 (5.7)
Females, *n* (%)	194,514 (41.9)	291 (19.2)	1055 (21.8)	9308 (34.9)	172,626 (42.6)	11234 (42.8)
Males, *n* (%)	267,771 (57.7)	1211 (80.0)	3414 (70.7)	16124 (60.4)	232,153 (57.3)	14869 (56.7)
Sex not known, *n* (%)	1799 (0.4)	12 (0.8)	361 (7.5)	1267 (4.7)	28 (0.0)	131 (0.5)
Total participants in studies that stratified sex, *n* (%)	17,102 (100)	0 (0.0)	365 (2.1)	1694 (9.9)	24 (0.1)	15,019 (87.8)
Included females in studies that stratified sex, *n* (%)	6288 (36.8)	0 (0.0)	30 (8.2)	4 (0.2)	10 (41.7)	6244 (41.6)
Included males in studies that stratified sex, *n* (%)	10,814 (63.2)	0 (0.0)	335 (91.8)	1690 (99.8)	14 (58.3)	8775 (58.4)
Age						
Mean age (range, SD)	56.1 (27–79, 8.8)	54.0 (54–54)	53.4 (36–74, 9.4)	59.3 (27–74, 9.0)	59.3 (27–74, 9.0)	61.0 (27–78, 9.2)

Categorical variables are presented as number (%). Continuous variables are presented as mean (range, standard deviation). Valid percentages are presented.

**Table 7 biomedicines-11-01435-t007:** Characteristics of the included angiotensin-converting enzyme inhibitors (ACEI) studies in the scoping review.

	Total	1979–1980	1981–1990	1991–2000	2001–2010	2011–2020
Number of studies						
Total, *n* (%)	671 (100)	7 (1.0)	130 (19.4)	294 (43.8)	206 (30.7)	34 (5.1)
Without mentioning sex distinction, *n* (%)	48 (7.2)	0 (0.0)	26 (20.0)	15 (5.1)	6 (2.9)	1 (2.9)
Without sex stratification, *n* (%)	564 (84.1)	4 (57.1)	85 (65.4)	259 (88.1)	190 (92.2)	26 (76.5)
With sex stratification, *n* (%)	28 (4.2)	1 (14.3)	3 (2.3)	11 (3.7)	6 (2.9)	7 (20.6)
Only including females, *n* (%)	0 (0.0)	0 (0.0)	0 (0.0)	0 (0.0)	0 (0.0)	0 (0.0)
Only including males, *n* (%)	31 (4.6)	2 (28.6)	16 (12.3)	9 (3.1)	4 (1.9)	0 (0.0)
Participants						
Total, *n* (%)	546,164 (100)	72 (0.0)	15,297 (2.8)	53,251 (9.8)	432,462 (79.2)	45,082 (8.3)
Females, *n* (%)	201,749 (36.9)	15 (20.8)	7138 (46.7)	15,572 (29.2)	164,447 (38.0)	14,577 (32.3)
Males, *n* (%)	341,789 (62.6)	57 (79.2)	7491 (49)	35,960 (67.5)	267,839 (61.9)	30,442 (67.5)
Sex not known, *n* (%)	2626 (0.5)	0 (0.0)	668 (4.4)	1719 (3.2)	176 (0.0)	63 (0.1)
Total participants in studies that stratified sex, *n* (%)	2738 (100)	31 (0.9)	256 (3.1)	1641 (91.6)	322 (2.8)	488 (1.6)
Included females in studies that stratified sex, *n* (%)	507 (18.5)	1 (3.2)	12 (4.7)	120 (7.3)	72 (22.4)	302 (61.9)
Included males in studies that stratified sex, *n* (%)	2231 (81.5)	30 (96.8)	244 (95.3)	1521 (92.7)	250 (77.6)	186 (38.1)
Age						
Mean age (range, SD)	58.3 (23–82, 8.5)	59.5 (54–65, 10.8)	58.0 (38–82, 9.2)	57.5 (23–81, 8.0)	59.3 (23–81, 8.7)	59.7 (47–71, 10.1)

Categorical variables are presented as number (%). Continuous variables are presented as mean (range, standard deviation). Valid percentages are presented.

**Table 8 biomedicines-11-01435-t008:** Characteristics of the included angiotensin receptor blockers (ARB) studies in the scoping review.

	Total	1984–1990	1991–2000	2001–2010	2011–2020
Number of studies					
Total, *n* (%)	324 (100)	2 (0.6)	28 (8.6)	210 (64.8)	84 (25.9)
Without mentioning sex distinction, *n* (%)	8 (2.5)	0 (0.0)	1 (3.6)	4 (1.9)	3 (3.6)
Without sex stratification, *n* (%)	295 (91.0)	2 (100)	24 (85.7)	192 (91.4)	77 (91.7)
With sex stratification, *n* (%)	14 (4.3)	0 (0.0)	0 (0.0)	10 (4.8)	4 (4.8)
Only including females, *n* (%)	0 (0.0)	0 (0.0)	0 (0.0)	0 (0.0)	0 (0.0)
Only including males, *n* (%)	7 (2.2)	0 (0.0)	3 (10.7)	4 (1.9)	0 (0.0)
Participants					
Total, *n* (%)	379,302 (100)	79 (0.0)	2642 (0.7)	300,251 (79.2)	76,330 (20.1)
Females, *n* (%)	160,182 (42.2)	21 (26.6)	638 (24.1)	124,292 (41.4)	35,231 (46.2)
Males, *n* (%)	209,658 (55.3)	58 (73.4)	1991 (75.4)	166,595 (55.5)	41,014 (53.7)
Sex not known, *n* (%)	9462 (2.5)	0 (0.0)	13 (0.5)	9364 (3.1)	85 (0.1)
Total participants in studies that stratified sex, *n* (%)	28,420 (100)	0 (0.0)	129 (0.5)	11,709 (41.2)	16,582 (58.3)
Included females in studies that stratified sex, *n* (%)	12,696 (44.7)	0 (0.0)	0 (0.0)	5717 (48.8)	6979 (42.1)
Included males in studies that stratified sex, *n* (%)	15,724 (55.3)	0 (0.0)	129 (100)	5992 (51.2)	9603 (57.9)
Age					
Mean age (range, SD)	59.7 (29–78, 8.9)	60.5 (57–64)	58.5 (42–73, 7.9)	59.6 (29–76, 8.6)	60.4 (39–78, 10.0)

Categorical variables are presented as number (%). Continuous variables are presented as mean (range, standard deviation). Valid percentages are presented.

## Data Availability

No individual patient data are included in this study. The search strategy and results of included papers are presented within the manuscript and are available from the corresponding author upon request.

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
