# Peer review of "The Representation of Females in Studies on Antihypertensive Medication over the Years: A Scoping Review"

_biomedicines, 2023, doi:10.3390/biomedicines11051435_

Round 1

Reviewer 1 Report

I was invited to revise the paper entitled "The representation of females in studies on antihypertensive medication over the years: a scoping review". It was a review aimed to evaluate the change in female partecipation and sex-stratified data in clinical studies for five major antihypertensive drugs.

Author performed a systematic review and tried to evaluate the trend of female enrollment across decades.

Major observations:

- Authors should report as supplementary materials all included study with relative characteristics;

- It is unknown if Authors included a single study several times: many RCT results are published in more srticles across years. It is unclear how Authors handled this point;

- In my opinion observational studies should be removed ;

- How did Authors considered RCT that evaluated more than one drug class?

- Why did Authors decided to perform the research only on two databases and not in 3?

The study is interesting and it focused on an important topic but many points should be clarified.

Author Response

Please see the attachment for rebuttal letter.

Reviewer 2 Report

The paper's topic is both relevant and exciting, as it addresses an important issue in clinical research: the representation of females in studies on antihypertensive medication. The underrepresentation of females in clinical trials has been a long-standing concern, and understanding how it has changed over time can help inform future research and policy.

 Hypertension is a prevalent condition with significant public health implications, and antihypertensive medications are widely prescribed to manage it. Ensuring that both males and females are adequately represented in clinical trials allows for a more accurate understanding of the efficacy and safety of these medications in different populations. Investigating sex-stratified data can help identify potential sex-related differences in treatment effects, ultimately leading to better, more personalized medical care.

This topic could have social and media relevance, given the public health implications. The findings could inform discussions on gender equity in medical research and raise awareness of the need for more inclusive and representative clinical trials. Additionally, the paper's conclusions could interest healthcare professionals, researchers, and policymakers, who may use this information to advocate for better research practices or to inform future guidelines and regulations.

 MINOR  ISSUES 

Material and methods

 The number of references reviewed is impressive. However, concerns arise regarding the feasibility of reviewing the titles and abstracts of 73,867 articles and assessing 15,130 full-text articles for eligibility. Given the time and resources required, the authors must provide a more detailed explanation of their methodology.

The methodology section of the paper should include the following:

 A clear description of the screening process and criteria used to filter the articles.

The number of reviewers involved and their roles in the study selection process.

The use of any specific software or tools to streamline the screening and data extraction process.

The timeline for the literature search, including when it began and how long it took.

A description of how the authors organized themselves to conduct the review, including any division of labor or collaboration among the team members.

Steps taken to ensure the quality and reliability of the data extraction process.

Addressing these concerns will strengthen the credibility of the study and provide readers with a better understanding of the rigorous approach employed by the authors.

The quality of the english is good. Authors  should correct minor typografic errors. 

Author Response

(The authors gave the same response as above.)

Round 2

Reviewer 1 Report

Now it can be accepted